# Early Weaning Impairs the Growth Performance of Hu Lambs Through Damaging Intestinal Morphology and Disrupting Serum Metabolite Homeostasis

**DOI:** 10.3390/ani15010113

**Published:** 2025-01-06

**Authors:** Haoyun Jiang, Haibo Wang, Haobin Jia, Yuhang Liu, Yue Pan, Xiaojun Zhong, Junhong Huo, Jinshun Zhan

**Affiliations:** 1Institute of Animal Husbandry and Veterinary, Jiangxi Academy of Agricultural Science, Nanchang 330200, China; jianghaoyun@jxaas.cn (H.J.); wanghaibo8815@163.com (H.W.); jiahaobin@jxaas.cn (H.J.); py13782525871@163.com (Y.P.); zhongcaoyangchu@163.com (X.Z.); 2College of Animal Science and Technology, Gansu Agricultural University, Lanzhou 730070, China; 3College of Animal Science and Veterinary Medicine, Tianjin Agricultural University, Tianjin 300384, China

**Keywords:** early weaning, growth performance, Hu lambs, ruminal development, serum metabolomics

## Abstract

Early weaning (EW) is an important breeding strategy for intensive sheep farming. Previous studies that have referred to the effect of EW on Hu lambs have mostly focused on growth performance and ruminal development. There are limited studies on gastrointestinal development and serum metabolite alteration in EW lambs. We thoroughly investigated the difference in growth performance, serum parameters, gastrointestinal development, and serum metabolites of Hu lambs weaned at 30 d (EW) and 45 d (conventional weaning) of age. The results identified that EW impaired the growth performance and intestinal morphology, while the ruminal development of Hu lambs was accelerated. Additionally, EW alters the concentrations of 5-HT, phenylalanine, tyrosine, and arachidonic acid. This research could supply a reference for formulating appropriate strategies to alleviate the EW stress for Hu lambs during transition from ewe rearing to artificial solid feed.

## 1. Introduction

In extensive sheep production systems based on natural pastures, lactation over long periods and the high mortality of sheep kids restrict sheep production development [1]. Thus, intensive meat sheep farms commonly use early, traditional, or late weaning after the lactation peak (1–2 days after birth) [2]. Early weaning (EW) is often carried out at 30 (D30) to 60 (D60) d of age, traditional weaning between D61 and D90, and late weaning after D90, but before natural weaning (no more than 12 months). Therefore, as a common breeding strategy, EW has been developed to shorten the breeding cycle and reduce production costs in intensive sheep farming [3]. EW methods can be performed abruptly by skipping milk feedings, progressive weaning, or two-stepped weaning. Among the methods, regardless of the age of the lambs, abrupt weaning is the most frequent method, which involves suddenly separating the ewe and offspring and avoiding contact. Progressive weaning involves reducing the ewe–lamb contact time and daily milk quantity in the transition phase. Two-stepped weaning limits the access to the ewes’ udder using a wire fence or other partitions before the final separation, and lambs have visual, auditory, or olfactory contact with their ewes without full physical contact (suckling) [3,4]. The proper EW strategy can accelerate the recovery of ewes’ physical condition and the adaptation of young ruminants to solid feed, which can further maximize the utilization of ewes and shorten the reproductive cycle [5]. However, the sudden termination of lactation, exogenous feed intervention, and abrupt nutrient changes can trigger psychological, physical, and immune stress responses in lambs, which limit the efficiency of EW [6]. Specifically, EW stress affects the structural and functional development of the intestinal tract, which leads to a decreased growth performance and higher diarrhea rate of lambs. A previous study showed that a 1.6% decrease occurred in the growth rate of lambs on the first day after weaning [7]. Additionally, it has been demonstrated that EW altered the glucose metabolism and decreased the immunoglobulin concentration and antibody titers, which affected the immune level of young calves [8]. However, other studies reported that EW at 4 weeks after birth can accelerate the morphological and physiological development of the rumen of lambs [9].

Metabolomics, as a powerful high-throughput biological analysis technique, integrates external (such as diets) and internal (such as genotypes) factors through analyzing metabolite alteration (lipids, sugars, and amino acids) [10,11]. Metabolomics has been widely used in research on metabolism changes in livestock [12], while studies focused on the blood metabolomic alteration of EW lambs under intensive feeding conditions are limited.

Hu is an excellent meat sheep breed that has a large-scale house farming history for over one thousand years in China due to its high reproductive performance (three–four sheep per litter) and stress resistance [13]. Nevertheless, the third and the fourth neonatal Hu lambs usually suffer from retarded growth and increased mortality, on account of weakling body condition and lack of colostrum from dams [14]. To decrease the high mortality rates, the milk replacer is generally fed to newborn Hu lambs in the intensive lambing industry. EW has the potential to reduce the artificial milk replacer rearing costs that are incurred in the rearing of Hu lambs and to increase the utilization rate of dams [15]. Thus, the effect of EW needs extensive research and attention. However, the studies that have referred to EW in lambs have mostly focused on growth performance [16,17], ruminal development [18,19], and ruminal microbial community [20,21]. Currently, there are few studies on the effect of EW on the physiological status, gastrointestinal development, and especially alteration in serum metabolites of Hu lambs. Therefore, we thoroughly investigated the effect of EW on the growth performance, serum parameters, gastrointestinal development, and serum metabolite composition of Hu lambs. The research could supply a reference for formulating appropriate strategies to alleviate the EW stress for Hu lambs during the transition from ewe rearing to artificial solid feed.

## 2. Materials and Methods

The experimental procedures involving Hu lambs were approved by the Animal Ethics Committee of Animal Husbandry and Veterinary, Jiangxi Academy of Agricultural Sciences, under permit No. 2010-JAAS-XM-01.

### 2.1. Animals, Diets, and Experimental Design

The animal trial was conducted in the Ganzhou Lvlinwan sheep breeding farm (Jiangxi, China). A total of 24 healthy male Hu lambs with similar birth weights (3.43 ± 0.103 kg) and good physical condition were selected from ewes that gave birth to twin males. All lambs were mixed in one sheepfold and initially ewe-reared to obtain sufficient colostrum intake for 24 to 48 h postpartum; uniform breeding conditions were maintained until 30 d of age (D30). Additionally, all lambs were provided pelleted starters twice daily (0800 and 1600 h) from D15 to D45. At the beginning of the weaning trial (D30), all 24 lambs were weighed and divided into two groups, with 12 lambs per group based on similar body weight (BW): the traditional weaning (control, CON) group was weaned at D45; the early weaning (EW) group was weaned at D30. Then, 12 lambs per group were placed in 6 indoor pens (1.5 × 1.5 m) to record the starter intake from D30 to D45. Each pen in CON group contained two lambs with their ewes; each pen in the EW group contained two lambs without their ewes. In brief, 12 lambs in the EW group were separated from their dams and abruptly weaned at D30, while the remaining 12 lambs in the CON group continued to be ewe-reared until D45. All lambs had ad libitum access to water with a drinker and had no access to the ewes’ diet throughout the entire trial.

Ewes were fed a total mixed ration consisting of 30% corn stover silage, 10% wheat bran 12% oat grass, 10% alfalfa, 8% barley straw, 5% cole stalk, 13% soybean residue, 9% corn, and 3% soybean meal. The starter diet used for supplementary feeding was formulated according to the feeding standard for mutton-producing sheep (NY/T 816-2004) [22]. The ingredients and nutrient compositions based on the dry matter basis of the starter diet are shown in Table 1.

### 2.2. Analysis of Growth Performance and Feed Intake of Hu Lambs

The BW of total 24 lambs (*n* = 12 per group) was recorded daily before morning feeding for calculating the average daily gain (ADG) from D30 (initial BW) to D45 (final BW). The ADG (*n* = 12 per group) was calculated as ADG = (final BW—initial BW)/15 d. Additionally, the amount of starter pellets offered and residual in each pen (*n* = 6 per group) were record daily from D30 to D45 to calculate the daily feed intake (ADFI).

### 2.3. Serum Sampling and Measures of Biochemical Parameters: Immunity and Inflammatory Status

A total of 6 lambs per group (1 lamb from each pen) were selected randomly to collect blood samples via jugular vein before the morning feeding at 30, 33, 36, and 45 d of age (D30, D33, D36, and D45); the samples were preserved in 10 mL plain vacuum tubes (Kangweishi Medical Technology Co., Ltd., Shijiazhuang, Hebei, China) and kept at 4 °C for 2 h. The blood samples were centrifuged at 3500 r/min for 10 min to obtain serum and then transferred to 1.5 mL cryovials and frozen at −20 °C for subsequent detection.

The serum biochemical parameters, including total protein (TP), total cholesterol (T-CHO), glucose (GLU), and blood urea nitrogen (BUN), were determined using the full-automatic serum biochemical analyzer (Nanjing Jiancheng Bioengineering Institute, Nanjing, Jiangsu, China). The serum concentrations of inflammatory and immune indices, including interleukin-1 (IL-1), interleukin-2 (IL-2), interleukin-6 (IL-6), tumor necrosis factor-(TNF-α), interferon-γ (IFN-γ), immunoglobulin A (IgA), and immunoglobulin G (IgG), were detected in the Hu lambs’ samples using commercial ultrasensitive ELISA kits according to the manufacturers’ protocols (Coibo Bio Technology, Shanghai, China).

### 2.4. Gastrointestinal Tissues Collection and Morphological Detection

After blood sampling on D45, six lambs per group were euthanized by captive bolt stunning and exsanguination. Then, the rumen, reticulum, omasum, and abomasum without connective tissue were removed from trunk and weighed and the rumen, jejunum, ileum, and duodenum were rinsed thoroughly with iced physiological saline and stored in 4% paraformaldehyde for tissue sectioning. After fixation, dehydration, trimness, and embedding, the blocks were stained with hematoxylin and eosin. The images of each sample were captured using a digital microscopy system. After morphological examination, the papillae height, width, and muscle layer thickness in the rumen, as well as the villus height (VH) and crypt depth (CD) of the small intestinal segment, were assessed using Image-Pro-Plus 8.0 software.

### 2.5. Serum Metabolite Extraction, LC-MS/MS Analysis, and Differential Metabolite Identification

The serum samples (100 μL) collected on D45 of Hu lambs were placed in the EP tubes and resuspended with 900 μL acetonitrile. Then, the mixtures were incubated using a low-temperature freezer at −20 °C for 10 min after vortexing and centrifuged at 12,000 × *g* for 10 min. Subsequently, the supernatants were transferred into fresh tubes for analysis using the LC-MS/MS analyses. UHPLC-MS/MS analyses were performed using the Vanquish UHPLC system (ThermoFisher, Waltham, MA, USA). The raw data were converted to mzXML format and imported to an in-house program with XCMS software 4.7 for peak identification, peak filtering, and peak alignment. Positive and negative ion modes were both used to detect metabolites. Additionally, orthogonal projections for the latent structure discriminant analyses (OPLS-DA) model were further performed to identify the biochemical patterns. The R2X, R2Y, Q2, and OPLS-DA score maps were used to evaluate the classification performance of the models. Next, the variable importance in the projection (VIP) values obtained by OPLS-DA and the *p*-values generated by a *t*-test of univariate statistical analysis were used to filtrate the significantly different metabolites. The thresholds for VIP ≥ 1.0 and *t*-test *p* < 0.05 were considered as differential metabolites between the CON and EW groups. The pathway enrichment analysis was completed using the Kyoto Encyclopedia of Genes and Genomes (KEGG) database on differential metabolites to compare and analyze the main biochemical metabolic pathways and signal transduction pathways involved in the differential metabolites between the CON and EW groups.

### 2.6. Statistical Analysis

The statistical difference between two treatments of growth performance, gastrointestinal weight, and morphology were evaluated using a Student’ s *t*-test using the SPSS v22.0 (SPSS Inc., Chicago, IL, USA). And these results were expressed as means ± standard error of the means (SEM). *p* < 0.05, 0.05 ≤ *p* < 0.10, and *p* > 0.05 were used as the criteria for significant difference, trend toward difference, and insignificant difference, respectively. Additionally, the blood parameters were analyzed using a two-way ANOVA model using the GLM procedure of SPSS according to the following model: Y_ijk_ = μ + α_i_ + β_j_ + αβ_ij_ + ε_ij_, where Y_ijk_ was the dependent variable, μ was the overall mean, α_i_ was the treatment effect, β_j_ was the age effect, and ε_ij_ was the residual error. The factors in the model included treatment, age, and their interaction. Differences were considered significant at *p* < 0.05. One-way ANOVA with Duncan’s multiple range tests for group comparison was performed only when significant differences in treatment, age, and their interaction were observed (*p* < 0.05). The multivariate and statistical analyses were performed using MetaboAnalyst 4.0 for metabolome data. The peaks extracted from serum samples was evaluated by OPLS-DA approaches, and differential metabolites with the FDR adjusted *p*-value < 0.05 and VIP > 1.0 were obtained using Student’ *t*-test. The different metabolic pathway enrichments were analyzed through online KEGG database (http://www.genome.jp/kegg/ (accessed on 15 September 2023)) annotation.

## 3. Results

### 3.1. Effect of EW on Growth Performance of Hu Lambs

As shown in Table 2, there was no significant difference in initial BW between the two groups (*p* > 0.05). Compared with the CON group, EW significantly decreased the final BW and ADG (*p* < 0.001) as well as ADFI in the first (*p* = 0.004) and second (*p* = 0.013) 5 days of the Hu lambs.

### 3.2. Effect of EW on Gastrointestinal Organic Weight of Hu Lambs

The gastrointestinal weights of Hu lambs at 45 d of age were shown in Table 3. The weights of rumen, reticulum, and omasum were observed to be higher in the EW group than those in the CON group (*p* < 0.05).

### 3.3. Effect of EW on the Morphology of Rumen and Small Intestine of Hu Lambs

As shown in Table 4, the significant differences in the morphological changes of the rumen and duodenum were observed between two groups at 45 d of age. EW significantly increased the papillae length (*p* < 0.05), while tended to improve the papillae width (*p* = 0.082) in the rumen of Hu lambs. Additionally, the villus height was observed to reduce, whereas the crypt depth was increased in the EW group compared with the CON group (*p* < 0.05).

### 3.4. Effect of EW on Serum Biochemical Parameters of Hu Lambs

The biochemical parameters levels in serum collected on D30, D33, D36, and D45 of Hu lambs between the CON and EW groups were shown in Table 5. Compared with the CON group, the concentration of GLU was shown to reduce in the EW group on D33, D36, and D45 (*p* < 0.001). The concentrations of GLU (*p* = 0.005), TP (*p* = 0.004), T-CHO (*p* < 0.001), and BUN (*p* < 0.001) were significantly affected by age. Additionally, both treatment and age interactively affected the levels of GLU (*p* = 0.001), TP (*p* = 0.041), and T-CHO (*p* = 0.022).

### 3.5. Effect of EW on Serum Inflammatory and Immune Status of Hu Lambs

The inflammatory and immune status in serum collected on D30, D33, D36, and D45 of Hu lambs between CON and EW groups were summarized in Table 6. Compared with CON group, the IL-6 content was lower in EW group on D45 (*p* = 0.018), while the levels of IgA (*p* = 0.027), IgG (*p* = 0.035), and IgM (*p* = 0.002) were all significantly higher in EW group on the four ages. Additionally, as for the age factor, the levels of IL-1, IL-2, IL-6, and TNF-α on D33 were the highest among the four ages (*p* < 0.001), while the levels of IgA (*p* = 0.001), IgG (*p* = 0.021), and IgM (*p* = 0.017) on D33 were the lowest among the four ages. Only IL-6 content was interactively affected by both treatment and age (*p* = 0.016).

### 3.6. Effect of EW on Serum Metabolites of Hu Lambs

To investigate the effect of EW on serum metabolites, non-targeted metabolomic analysis was performed on serum samples of two groups lambs in this study. To obtain reliable and high-quality metabolomics data, quality control was conducted first on QC samples, and the results showed good repeatability in the QC samples (Figure 1). After recognition, filtering, and alignment processing in the peak, a total of 901 and 981 metabolites were identified in the positive and negative ion modes, respectively. The OPLS-DA model was used to further evaluate the reliability of the metabolomics data (Figure 1A,B). The OPLS-DA models showed excellent explanatory and predictive abilities, with R2Y and Q2 values of 0.926 and 0.817 in the positive ion mode (Figure 1C) and R2Y and Q2 values of 0.975 and 0.88 in the negative ion mode (Figure 1D); the *Y*-axis intercept in the permutation test results was less than 0. The OPLS-DA model results were not overfitted, and had good predictive ability, indicating the metabolome data have high reliability. Additionally, the OPLS-DA plots showed a clearly distinguishable separation in the metabolites between the CON and EW groups.

As shown in Figure 2A,B, a total of 59 and 55 differential metabolites in the positive and negative ion modes were found in serum of Hu lambs. Compared with the CON group, the contents of serotonin (5-hydroxytryptamine, 5-HT) and arachidonic acid were significantly increased, while indole, L-phenylalanine, L-tyrosine, and L-glutamic acid were reduced in the EW group (*p* < 0.05) (Figure 2C,D). The top 20 KEGG pathway analyses showed that the differential metabolites of the positive and negative ion modes were jointly enriched in protein digestion and absorption, phenylalanine, tyrosine and tryptophan biosynthesis, and inflammatory mediator regulation of TRP channels (Figure 3).

## 4. Discussion

EW can accelerate the ewes breeding circle in the Hu sheep industry. Therefore, it is traditionally believed that switching lambs from milk to solid feed as early as possible can increase profits [2]. Additionally, it is generally acknowledged that the consumption and fermentation of solid feed before weaning will promote rumen development and function [23]. However, the sudden separation from ewes and the complete replacement of milk with solid food always contribute to physiological and nutritional pressure on EW lambs [5,24]. Therefore, our current study explored the effect of EW on growth performance, serum parameters and metabolites, and gastrointestinal development of Hu lambs in order to minimize the negative impacts of EW.

### 4.1. EW Impairs Growth Performance of Hu Lambs

EW is always accompanied by intestinal stress and diarrhea, further leading to the reduction in growth the performance of lambs [17]. It has recently been reported that BW, ADG, and starter intake of lambs is decreased in EW groups (weaning on D21) in comparison those in CON groups (weaning on D49) [16,17]. Chai et al. [25] also showed that lambs with EW on D30 showed decreased ADG in comparison with ewe-reared lambs within 10 days post-weaning. Additionally, previous research has reported that EW decreases nutrient intake and digestion, including crude protein (CP) and ether extract (EE) [26]. In our present study, compared with lambs in the CON group (weaned on D45), the final BW, ADG, and ADFI of Hu lamb in the EW group in the first and second 5 days were significantly decreased. In line with the literature [2,27], our results confirmed the known effect of EW on growth performance. The reduced ADG of lambs in the EW group was related to the lack of milk meal intake, which only came from the starter nutrient source. Additionally, abrupt stress in feeding pattern and psychological affection always causes a lower growth performance. The decreased ADFI in the EW group showed no significant difference in the third 5 days. We speculated the lambs adapted to the starter pellets intake without their ewes and suckling milk. The EW lambs exhibited a lower final BW than their traditional weaning counterparts, again indicating that EW stress has a negative impact on the immature gastrointestinal tract, despite solid feed intervention from the infancy period, leading to a long-term negative effect on growth.

### 4.2. EW Impacts Rumen and Small Intestine Development in Hu Lambs

The rumen, as a little pouch-like tissue with a thin lining, is immature in newborn ruminant, and is thus unable to digest and absorb the nutrients of a solid diet. In artificially reared ruminants, the preweaning development of the ruminal absorbed function is crucial in rearing ruminant cubs with the adaptation of the transition from milk to solid diets, effecting the growth performance of lambs [28]. It has been revealed that smaller ruminal development was found in goats’ kids solely fed with the milk from dams [29]. The solid feed supplementation promoted the rumen papillae growth and increased the ruminal volatile fatty acid concentrations [30]. Furthermore, longitudinal morphology development research on lambs has determined that rumen and omasum weight present an abrupt increase following solid feed intervention after EW [29]. Compelling evidence indicates that lambs in EW groups have significantly higher rumen papillae epithelium length than those in CON group [9,31]. This is consistent with the findings of the present study; we observed a significant increase in papillae length, rumen weight, and rumen organ index in the EW group in comparison with the CON group on D45. Thus, it is reasonable to speculate that solid feed consumption as an initiating agent might promote rumen epithelium development in Hu lambs [32].

Studies on EW in ruminants have mainly focused on growth performance, blood biochemical parameters, and ruminal development. However, relatively few studies have been conducted on the impact of EW on the intestinal growth of ruminants. It has been widely demonstrated that the small intestine plays a crucial role in nutrient absorption, the immunity barrier, and the postnatal growth of lambs [33,34]. Unlike early solid feed intervention in the first few weeks, which promotes ruminal development, EW always impairs the digestive function of the small intestine of calves [27]. Previous studies have shown that EW damages jejunum and ileum development and leads to incomplete mucosal villi in lambs [35]. In the present study, EW decreased the villus height; meanwhile, it appeared to lead to an increase in the crypt depth in the duodenum on D45. Additionally, similarly to our data, earlier studies have reported that EW decreases the villi width [16] and alters barrier function in the small intestines of lambs [36]. The EW stress triggered by the transition from milk to solid diets mainly impaired the duodenal development; this might be because the duodenum is the first intestinal segment that is in contact with the chyme.

### 4.3. EW Disrupts the Serum Biochemical Parameters of Hu Lambs

Serum biochemical parameters are helpful indicators of the physiological, nutritive, and pathological status of animals. Blood TP concentration can be used as an indicator of milk and dietary protein utilization efficiency, and a decrease in TP content leads to a decrease in diet protein utilization efficiency in young lambs [37]. Consistent with a previous study [14], the results of the current study show that EW decreased serum TP concentration on D33 and D36 in lambs, which suggested that EW reduced the dietary protein utilization and further resulted in protein anabolism deficiency, whereas other research reported EW did not affect TP concentration [25]. A previous study has shown that EW reduced the dietary CP and EE ingestion and digestion, which may be associated with impaired intestinal amino acid absorption and utilization rates [26]. Furthermore, EW lambs showed a lower plasma GLU level on D33, D36, and D45. Previous evidence suggests that EW causes impairment in dietary GLU digestibility throughout life and results in a lower plasma GLU level. Therefore, we speculated that EW might lead to an impaired intestinal GLU and TP absorption efficiency among Hu lambs.

### 4.4. EW Alters Serum Immune and Inflammatory Response of Hu Lambs

Weaning stress activates immune and inflammatory responses and leads to the secretion of a large number of proinflammatory cytokine biomarkers; the overproduction cytokine is always accompanied by intestinal injury and dysfunction [38]. A previous study reported that EW can provoke the synthesis and secretion of proinflammatory cytokines such as TNF-α, IL-1, and IL-6, further activating an inflammatory reaction [39]. In the present study, our results showed that EW significantly increased the contents of TNF-α and IL-6 on D30, while decreasing the level of IL-6 on D45. Obviously, the higher proinflammatory cytokines on D30 were induced by EW. However, due to the short-term acute characteristics of inflammatory responses, a “cytokine storm” does not last for a long time after weaning [17]. And our results proved this: the levels of TNF-α, IL-1, and IL-6 on D30 were higher in the EW group, but showed no difference, even though the levels were lower, on D36 and D45. It is widely recognized that newborn animals have inadequate immunity, and their survival rates only rely on passive immunity transmitted especially through immunoglobulin from the mother to the offspring through breast milk ingestion [40]. IgG has been found to account for 75% of total immunoglobulins; jointly, IgA plays a prime role in the primary immune response to prevent pathogen and viral invasion [41]. In our results, the concentrations of IgA and IgG were found to be higher in the EW group than in the CON group on D45. Consistent with our results, Li et al. [42] also reported that EW increased the concentrations of IgA and IgM after weaning (D31), indicating that the change might be due to the synthesis of endogenous immunoglobulins among lambs without milk-derived antibodies. Nevertheless, further longer-term research is required to acquire the relevant immune index support.

### 4.5. Metabolite Composition Differed Between CON and EW Groups

Up to now, there has been little study focused on the effect of EW on the blood metabolome of Hu lambs; it is necessary to formulate a nutritional intervention strategy based on certain metabolites to prevent weaning stress. We found a total of 59 positive ion modes and 55 negative ion modes for differential metabolites in the serum of the lambs in the CON and EW groups; of these, 5-HT drew our attention. 5-HT has always been considered to be an important signaling molecule in the intestine, promoting inflammation and acting as a nutritional factor [43]. Previous studies have also showed that enterochromaffin cells secret a large amount of 5-HT in the intestinal cavity under various physiological stimuli [44]. In this study, the level of 5-HT in the serum of the lambs in the EW group was higher than that of the lambs in the CON group; this might have been triggered by the sudden replacement of milk with solid food. EW always leads to enterocyte adaptive stress reactions, which cause intestinal cells to secrete a large amount of 5-HT. The phenotypes of the damaged intestinal VH and higher 5-HT concentration jointly indicate that intestinal stress induced by EW could cause intestinal inflammatory injury. EW, as a highly stressful event, increases the demand for essential amino acids (EAAs) in dietary protein ingestion to synthesize acute phase proteins [45]. Phenylalanine, as an EAA [46], is mainly metabolized and translated into tyrosine under normal physiological conditions in the liver and other tissues [47]. The concentrations of phenylalanine, tryptophan, and tyrosine in the liver [48] as well as glutamine in the plasma and jejunal fluid [49] of the weaned piglets were observed to significantly reduce. It is worth noting that the contents of L-phenylalanine, L-tyrosine, and L-glutamic acid were detected to significantly decrease in the serum of EW lambs in the current study. It has been reported that the addition of glutamic acid could increase growth performance through improving the mucosal integrity of the small intestine in EW piglets [50]. Therefore, supplementation with EAAs might contribute to improving the growth and gut development of Hu lambs after EW. Based on the above results, we speculated that EW stress could cause amino acid metabolism disturbance in serum, especially phenylalanine and tyrosine balance, in Hu lambs.

## 5. Conclusions

Taken together, the current results identified that EW at 30 d of age decreased the growth performance (ADG and ADFI) of Hu lambs within two weeks post-weaning; this effect might be associated with impaired intestinal morphology, especially duodenal villus height, and glucose metabolism. The serum metabolomics analysis revealed that EW altered the concentrations of intestine-derived 5-HT, phenylalanine, tyrosine, and arachidonic acid, which could serve as potential regulatory targets for modulating the health of EW Hu lambs. This study provided new insights for alleviating EW stress through nutritional strategies based on the alteration of serum biochemicals and metabolites in Hu lambs (Figure 4).

## Figures and Tables

**Figure 1 animals-15-00113-f001:**
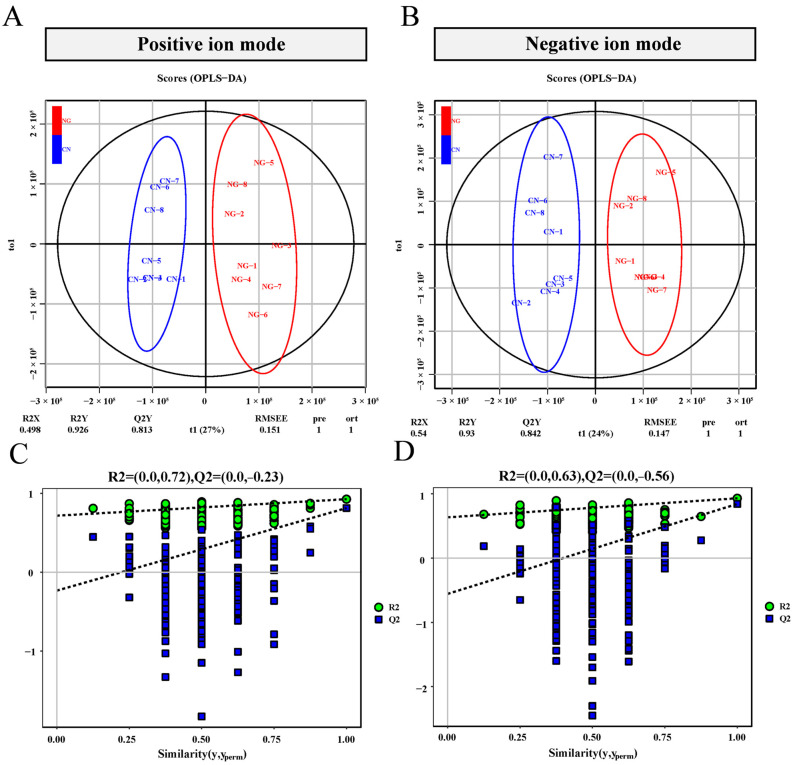
The orthogonal projections to latent structures discriminant analyses (OPLS-DA), and permutation score plots based on LC-MS between CON and EW groups. OPLS-DA score derived from the untargeted LC-MS profiles in the CON (blue dots) and EW (red dots) groups in the positive (**A**) and negative ion modes (**B**). Permutation score plot in the positive (**C**) and negative (**D**) ion modes. CON = control group, lambs were weaned at 45 d of age; EW = early-weaning group, lambs were weaned at 30 d of age.

**Figure 2 animals-15-00113-f002:**
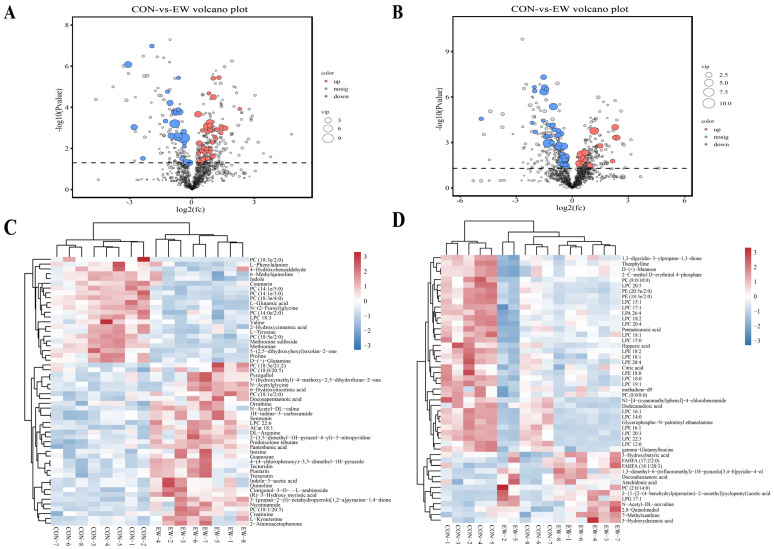
Differentially accumulating metabolites between CON and EW groups. Volcano maps of serum metabolites in positive (**A**) and negative (**B**) ion modes. Heatmap of 59 differential metabolites in positive (**C**) and 50 differential metabolites in negative (**D**) ion modes. CON = control group, lambs were weaned at 45 d of age; EW = early-weaning group, lambs were weaned at 30 d of age.

**Figure 3 animals-15-00113-f003:**
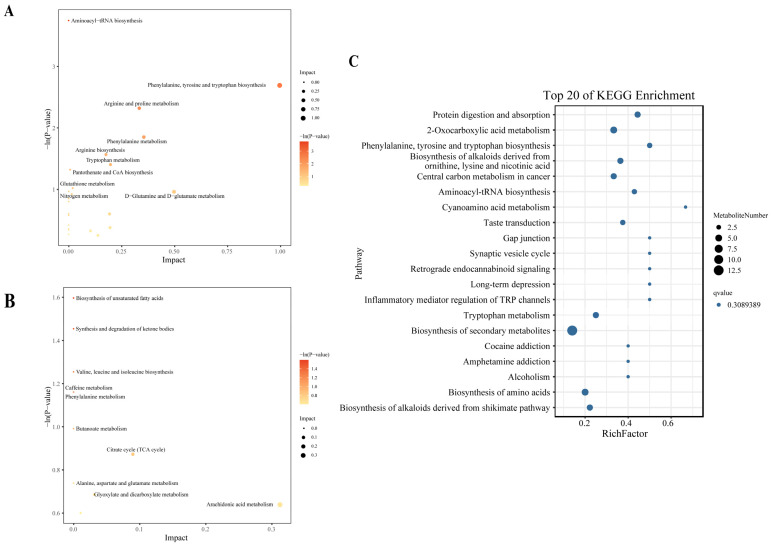
Differentially pathway enrichment analysis between CON and EW groups. Key metabolic pathways analysis of differential metabolites in positive (**A**) and negative (**B**) ion modes. The top 20 KEGG pathways displayed enrichment in differential metabolites between CON and EW group (**C**). CON = control group, lambs were weaned at 45 d of age; EW = early-weaning group, lambs were weaned at 30 d of age.

**Figure 4 animals-15-00113-f004:**
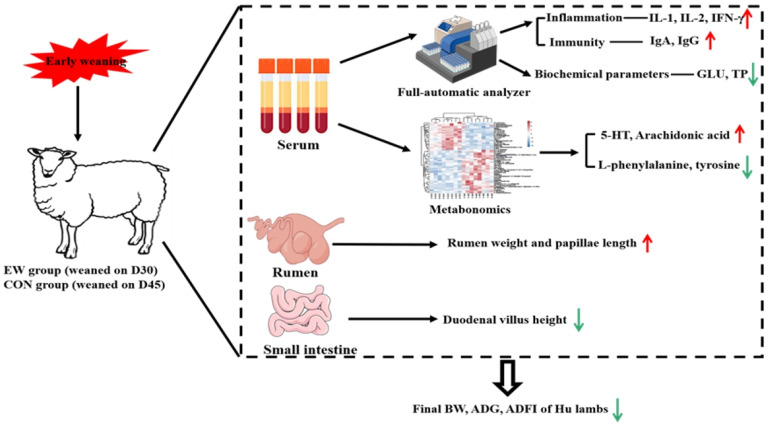
The study design and schematic diagram of the effect of early weaning (EW) on the growth performance, gastrointestinal development, serum parameters, and metabolomics of Hu sheep lambs.

**Table 1 animals-15-00113-t001:** Ingredients and nutrient compositions of the starter diet (dry matter basis, %).

Ingredients	Contents	Nutrient Compositions	Contents
Corn	63.50	DE (MJ/kg) ^2^	12.29
Soybean meal	16.50	CP	18.85
Cottonseed meal	6.00	Ash	7.35
Rapeseed meal	5.00	EE	3.44
Dried corn with alcohol grains	1.20	NDF	28.24
Corn husk	4.32	ADF	12.54
Stone powder	1.18	Calcium	1.49
Calcium hydrogen phosphate	1.00	Phosphorus	0.73
Sodium chloride	0.30		
Premix ^1^	1.00		
Total	100.00		

ADF = acid detergent fiber; CP = crude protein; DE = digestible energy; EE = ether extract; NDF = neutral detergent fiber. ^1^ Premix is provided for per kilogram of lamb feed: vitamin A 28,000 IU, vitamin D 34,000 IU, vitamin E 16 IU, vitamin K 30.2 mg, niacin 16 mg, copper 16 mg, zinc 120 mg, selenium 0.3 mg, iron 36 mg, iodine 0.6 mg, and manganese 5.6 mg. ^2^ DE is calculated according to the composition of raw materials, and all other nutrient levels are measured values.

**Table 2 animals-15-00113-t002:** Effect of early weaning (EW) on growth performance (*n* = 12) and feed intake (*n* = 6) of Hu lambs.

Items	CON	EW	*p*-Value
Initial BW (kg)	10.06 ± 0.58	9.38 ± 0.78	0.487
Final BW (kg)	14.72 ± 0.71	10.60 ± 0.66 *	<0.001
ADG (g/d)	310.47 ± 18.24	81.86 ± 12.78 *	<0.001
ADFI (g/d)			
30–35 days	201.25 ± 11.48	128.80 ± 11.98	0.004
36–40 days	274.55 ± 10.84	236.03 ± 7.69	0.013
41–45 days	269.75 ± 12.88	262. 35 ± 9.82	0.677

Data are presented as means ± SEM. * *p* < 0.05. Initial BW, body weight at 30 d of age; Final BW, body weight at 45 d of age; ADG, average daily gain; ADFI, average daily feed intake; CON = control group, lambs were weaned at 45 d of age; EW = early-weaning group, lambs were weaned at 30 d of age.

**Table 3 animals-15-00113-t003:** Effect of early weaning (EW) on gastrointestinal organ index of Hu lambs at 45 d of age (*n* = 6).

Items	CON	EW	*p*-Value
Rumen fluid pH	7.92 ± 0.21	7.84 ± 0.17	0.775
Rumen weight (g)	94.95 ± 10.51	177.15 ± 12.11 *	0.002
Reticulum weight (g)	14.20 ± 0.91	28.41 ± 3.43 *	0.007
Omasum weight (g)	6.15 ± 0.26	10.68 ± 1.33 *	0.016
Abomasum weight (g)	49.28 ± 5.28	38.95 ± 2.58	0.129
Small intestinal length (m)	17.32 ± 0.35	15.69 ± 0.88	0.139
Small intestinal weight (g)	286.13 ± 17.07	249.81 ± 15.05	0.162
Large intestinal length (m)	2.41 ± 0.13	2.67 ± 0.18	0.241
Large intestinal weight (g)	190.45 ± 23.98	176.12 ± 16.98	0.643

Data are presented as means ± SEM. * *p* < 0.05. CON = control group, lambs were weaned at 45 d of age; EW = early-weaning group, lambs were weaned at 30 d of age.

**Table 4 animals-15-00113-t004:** Effect of early weaning (EW) on ruminal and intestinal morphology of Hu lambs at 45 d of age (*n* = 6).

Items	CON	EW	*p*-Value
Rumen			
Papillae length (μm)	1061.77 ± 111.89	1587.33 ± 202.78 *	0.035
Papillae width (μm)	434.71 ± 19.14	528.79 ± 22.31	0.082
Muscle layer thickness (μm)	1034.5 ± 73.47	1097.59 ± 114.8	0.663
Papillae length to width ratio	2.53 ± 0.41	2.96 ± 0.41	0.492
Duodenum			
Villus height (μm)	550.51 ± 22.61	439.99 ± 24.24 *	0.016
Crypt depth (μm)	172.67 ± 13.88	226.10 ± 7.36 *	0.014
Villus height to crypt depth ratio	3.26 ± 0.32	1.95 ± 0.09	0.082
Jejunum			
Villus height (μm)	409.41 ± 50.63	385.83 ± 21.96	0.655
Crypt depth (μm)	157.29 ± 11.11	151.80 ± 10.75	0.823
Villus height to crypt depth ratio	3.21 ± 0.48	2.63 ± 0.27	0.337
Ileum			
Villus height (μm)	481.29 ± 22.73	542.34 ± 69.03	0.433
Crypt depth (μm)	142.69 ± 13.28	151.59 ± 15.03	0.764
Villus height to crypt depth ratio	3.48 ± 0.41	3.82 ± 0.59	0.645

Data are presented as means ± SEM. * *p* < 0.05. CON = control group, lambs were weaned at 45 d of age; EW = early-weaning group, lambs were weaned at 30 d of age.

**Table 5 animals-15-00113-t005:** Effect of early weaning (EW) on the biochemical parameters in serum collected at 30, 33, 36, and 45 d of age (D30, D33, D36, and D45) of Hu lambs (*n* = 6).

Items	Treatment	Day of Age				SEM	*p*-Value		
		30	33	36	45		Treatment	Age	Interaction
GLU, mmol/L	CON	5.74 ^b^	8.62 ^a^	10.00 ^a^	8.64 ^a^	0.272	<0.001	0.005	0.006
	EW	6.13 ^b^	5.70 ^b^	6.29 ^b^	6.28 ^b^				
TP, mg/mL	CON	83.53 ^abcd^	110.50 ^a^	91.69 ^abc^	64.33 ^cd^	3.818	0.086	0.004	0.041
	EW	94.99 ^ab^	85.92 ^abc^	54.96 ^d^	67.67 ^bcd^				
T-CHO, mmol/L	CON	4.42 ^bcde^	2.21 ^e^	5.09 ^bcd^	6.72 ^ab^	0.383	0.212	<0.001	0.022
	EW	2.97 ^de^	3.54 ^cde^	8.89 ^a^	6.11 ^bc^				
BUN, mmol/L	CON	2.61 ^cde^	3.26 ^bcde^	3.92 ^bcd^	4.84 ^b^	0.997	0.578	<0.001	0.127
	EW	2.07 ^e^	2.49 ^de^	4.36 ^bc^	6.63 ^a^				

^a–e^ Means with different superscripts differ significantly within two rows (eight values including age and treatment; *p* < 0.05). GLU = glucose; T-CHO = total cholesterol; TP = total protein; BUN = blood urea nitrogen. CON = control group, lambs were weaned at 45 d of age; EW = early-weaning group, lambs were weaned at 30 d of age.

**Table 6 animals-15-00113-t006:** Effect of early weaning (EW) on the inflammatory and immune indices in serum collected at 30, 33, 36, and 45 d of age (D30, D33, D36, and D45) of Hu lambs (*n* = 6).

Items	Treatment	Day of Age				SEM	*p*-Value		
		30	33	36	45		Treatment	Age	Interaction
IL-1, pg/mL	CON	602.95 ^bc^	740.74 ^a^	604.78 ^bc^	638.84 ^b^	15.582	0.317	<0.001	0.056
	EW	637.73 ^b^	790.81 ^a^	528.88 ^c^	533.81 ^bc^				
IL-2, pg/mL	CON	576.70 ^bc^	801.64 ^a^	643.33 ^b^	602.90 ^bc^	17.365	0.137	<0.001	0.169
	EW	634.57 ^b^	759.77 ^a^	554.51 ^bc^	513.45 ^c^				
IL-6, pg/mL	CON	129.28 ^cde^	162.53 ^a^	135.76 ^bcd^	141.14 ^bc^	4.616	0.018	<0.001	0.016
	EW	140.73 ^bc^	152.47 ^ab^	119.84 ^de^	115.05 ^e^				
TNF-α, pg/mL	CON	161.42 ^bcd^	197.59 ^ab^	164.89 ^bcd^	165.33 ^bcd^	5.056	0.057	<0.001	0.128
	EW	172.53 ^bc^	179.74 ^a^	152.14 ^cd^	144.89 ^d^				
IFN-γ, pg/mL	CON	554.02 ^a^	439.50 ^bc^	475.20 ^abc^	463.65 ^abc^	12.883	0.489	0.008	0.118
	EW	515.75 ^ab^	406.41 ^c^	506.69 ^abc^	569.00 ^a^				
IgA, pg/mL	CON	165.20 ^ab^	118.75 ^c^	159.14 ^ab^	155.22 ^ab^	4.742	0.027	0.001	0.661
	EW	177.62 ^ab^	139.73 ^bc^	166.60 ^ab^	190.62 ^a^				
IgG, pg/mL	CON	1432.19 ^ab^	1170.21 ^b^	1453.62 ^a^	1305.19 ^ab^	33.458	0.035	0.021	0.767
	EW	1522.15 ^a^	1303.41 ^ab^	1526.23 ^a^	1549.32 ^a^				
IgM, pg/mL	CON	74.27 ^bc^	67.02 ^c^	69.73 ^c^	69.80 ^c^	2.379	0.002	0.017	0.071
	EW	76.22 ^bc^	67.97 ^c^	80.62 ^ab^	86.31 ^a^				

^a–e^ Means with different superscripts differ significantly within two rows (eight values including age and treatment; *p* < 0.05). IL-1 = interleukin-1; IL-2 = interleukin-2; IL-6 = interleukin-6; TNF-α = tumor necrosis factor-α; IFN-γ = interferon-γ; IgA = immunoglobulin A; IgG = immunoglobulin G; IgM = immunoglobulin M. CON = control group, lambs were weaned at 45 d of age; EW = early-weaning group, lambs were weaned at 30 d of age.

## Data Availability

The raw data supporting the conclusions of this study will be made available by the authors, without undue reservation.

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
