# Peer review of "Early Weaning Impairs the Growth Performance of Hu Lambs Through Damaging Intestinal Morphology and Disrupting Serum Metabolite Homeostasis"

_animals, 2025, doi:10.3390/ani15010113_

Round 1

Reviewer 1 Report (Previous Reviewer 1)

Comments and Suggestions for Authors

I am satisfied with the corrections made

Author Response

Comment:I am satisfied with the corrections made

Response:Thank you very much for the reviewers’ effort on our manuscript!

Reviewer 2 Report (Previous Reviewer 2)

Comments and Suggestions for Authors

The authors followed the reviewer recommendations.

Figure 2 is still not readable. Please improve it.

The conclusions have to be improved. Rewrite them more clearly and detailed.

Author Response

Comment:The authors followed the reviewer recommendations.

Figure 2 is still not readable. Please improve it.

The conclusions have to be improved. Rewrite them more clearly and detailed.

Response:Thank you very much for the reviewers’ effort on our manuscript!We have provided Figures 2 in the highest resolution and made the font bigger for reading. The conclusions have been revised  detailed based on specific index as “Taken together, the current results identified that EW at 30 d of age decreased the growth performance (ADG and ADFI) of Hu lambs within post-weaning two weeks, which might be associated with impaired intestinal morphology especially duodenal villus height and glucose metabolism. Serum metabolomics analysis revealed that EW altered the concentrations of intestine-derived 5-HT, phenylalanine, tyrosine, and arachidonic acid, which could be served as potential regulatory targets for modulating the health of EW Hu lambs. This study provided new insights for alleviating EW stress through nutritional strategies based on the alteration of serum biochemicals and metabolites in Hu lambs”. 

Reviewer 3 Report (Previous Reviewer 3)

Comments and Suggestions for Authors

Dear Authors,

Thank you very much for your answers and efforts to revise the manuscript!

I agree with your modifications and accept all of them. 

There is only one recommendation and comment from my side. I can not find the two new literatures in the list of references, according to the early weaning. Please rivise this in the text.

Author Response

Comments: Dear Authors,

Thank you very much for your answers and efforts to revise the manuscript!

I agree with your modifications and accept all of them. 

There is only one recommendation and comment from my side. I can not find the two new literatures in the list of references, according to the early weaning. Please revise this in the text.

Response:Thank you very much for your effort on our manuscript!We have deleted the literature and replaced with other reference as "Zamuner, F.; Leury B.J.; DiGiacomo, K. Review: Feeding strategies for rearing replacement dairy goats - from birth to kidding. Animal. 2023, 17, 100853, doi:10.1016/j.animal.2023.100853".

This manuscript is a resubmission of an earlier submission. The following is a list of the peer review reports and author responses from that submission.

Round 1

Reviewer 1 Report

Comments and Suggestions for Authors

Lines 56-58: this sentence could be omitted or to be adapted to the final sentence of the introduction

How was determined the minimum sample size of animals for each group? Why was selected the pen as experimental unit and not the animals?

How often the health status of the animals was evaluated? Were any diseases such as diarrhea detected throughout the experiment? This information is important because may significantly affect the parameters evaluated

Line 117: what kind of coagulant? Usually, plain tubes are used for serum

Line 159: how was the normality checked? In my opinion, a model combining group and time should have been used for the evaluation of blood parameters that were repeatedly measured

Line 288: the immaturity of gastrointestinal tract despite solid feed consumption since D15 could be a cause for this condition?

Line 302-304: This sentence is somehow confusing, please rephrase

Reviewer 2 Report

Comments and Suggestions for Authors

Materials and methods - line 87 - please mention the mean of the birth weight of the lambs. Also for lines 90-91 - mention the mean of the lambs weight.

The experimental design is not clearly presented. Please mention if all the lambs were separated by the ewes. In which conditions? You said that all the lambs were feed twice with basic diet. Even those who were weaned at D45? Present in detail the feeding differences between D45 si D30. When you refere to the D45 you have to take into account also the milk meal nutrient content, for the period 30-45 days. In my opinion the experimental design it's not clear and have to be rewrite.

Figure 3, 4 and 5 are to small and are not readable.

Reviewer 3 Report

Comments and Suggestions for Authors

General comments of the reviewer:

The manuscript is  dealing with a current topic. The structure of the manuscript and the illustrations used in the text were compiled with scientific precision (see detailes below). The literatures used are relevant, and the methods used were appropriate. Refining of the keeping system and technology is recommended. All of the comments are listed in the attachment as "Comments".

Detailed comments and questions of the reviewer:

- It is not clear from the text of the manuscript what the purpose and benefit of early weaning are in Hu sheep keeping. It would be useful to give a detailed description of the different-time weaning methods (early, super-early, immediate, traditional), comparing their purposees and significance in the country's sheep sector (e.g. in chapter Introduction). This detailed description could support the basic objective of the manuscript, according to which it is important to investigate the effects of the early weaning on Hu sheeps' growth performance.

Resolution of Figure 3, 4 and 5 should be increased for better understanding.

Other comments are in the attached version of the manuscript.
